# Palm Tocotrienol Preserves Trabecular Osteocyte Indices and Modulates the Expression of Osteocyte Markers in Ovariectomized Rats

**DOI:** 10.3390/biomedicines13051220

**Published:** 2025-05-18

**Authors:** Sophia Ogechi Ekeuku, Shafiq Zikry Zarir, Anis Nazira Razali, Syamima Mohamad Zaidi, Noor Halinah Mohamed Ali Jinnah, Muhamed Lahtif Nor Muhamad, Sok Kuan Wong, Kok-Yong Chin

**Affiliations:** Department of Pharmacology, Faculty of Medicine, Universiti Kebangsaan Malaysia, Cheras, Kuala Lumpur 56000, Malaysia; virgosapphire2088@yahoo.com (S.O.E.); a183600@siswa.ukm.edu.my (S.Z.Z.); a180856@siswa.ukm.edu.my (A.N.R.); a184220@siswa.ukm.edu.my (S.M.Z.); a181430@siswa.ukm.edu.my (N.H.M.A.J.); muhamed_lahtif@yahoo.com (M.L.N.M.); sokkuan@ukm.edu.my (S.K.W.)

**Keywords:** dentin matrix protein-1, menopause, estrogen deficiency, sclerostin, vitamin E

## Abstract

**Background/Objective:** Palm tocotrienol has bone-protective properties in animal models, yet its underlying mechanism remains unclear. Given osteocytes’ role in bone homeostasis, this research aimed to investigate the effects of palm tocotrienol on the quantity of osteocytes and the expression of osteocyte-specific markers in ovariectomized rats. **Methods:** Adult female rats (Sprague Dawley; three-month-old; *n* = 6/group) were randomly divided into baseline, sham control, ovariectomized control, unemulsified palm tocotrienol (UPT), emulsified palm tocotrienol (EPT), and positive control. The baseline group was euthanized without intervention, whereas the sham group underwent a laparotomy procedure in which the ovaries were not excised. The other groups underwent bilateral removal of the ovaries and subsequently received UPT (100 mg/kg/day, 50% vitamin E), EPT (100 mg/kg/day, 25% vitamin E), or a combination of glucosamine sulfate (250 mg/kg/day) and calcium carbonate (1% in drinking water). Control groups were induced with similar gavage stress with olive oil. After 10 weeks, all rats were sacrificed for bone and serum analysis. **Results:** UPT and EPT significantly increased trabecular osteocyte and total lacunae numbers (*p* < 0.05 versus ovariectomized control). Both treatments significantly reduced mRNA expression levels of dentin matrix protein-1 (*p* < 0.05 versus ovariectomized control), whereas sclerostin mRNA expression was unchanged (*p* > 0.05 versus ovariectomized control). However, neither UPT nor EPT improved circulating or skeletal redox status (*p* > 0.05 versus ovariectomized control). **Conclusions:** Palm tocotrienol may support bone health by preserving the quantity of trabecular osteocytes and modulating osteocyte-mediated bone remodeling. Further research is required to elucidate its precise mechanisms.

## 1. Introduction

Osteoporosis, marked by a decline in bone mass and structural integrity, is a major cause of skeletal fragility and fractures [1]. It is a burgeoning public health issue because of the aging world population [2]. Postmenopausal women are particularly vulnerable to osteoporosis [3]. The global osteoporosis prevalence was estimated to be 18.3% and higher in women (23.1%) than in men (11.7%) [4]. This phenomenon is attributed to estrogen deficiency after menopause, skewing the bone remodeling process toward resorption and leading to bone loss [5].

Bone remodeling is a precisely regulated process encompassing bone resorption by osteoclasts and bone formation by osteoblasts [6]. The role of osteocytes (terminally differentiated osteoblasts entombed in a mineralized matrix) has attracted interest [7]. They regulate the activities of osteoblasts and osteoclasts through mechanosensing [8]. For instance, osteocytes influence osteoclastogenesis by secreting receptor activator of nuclear factor kappa-B ligand (RANKL). They secrete bone formation inhibitors, such as sclerostin (SOST) and Dickkopf-1 (DKK1), which suppress Wnt signaling [9], and dentin matrix protein 1 (DMP1), which is an extracellular matrix pro-peptide and a component of the small integrin-binding ligand N-linked glycoprotein family associated with bone matrix mineralization [10]. Of note, Wnt signaling is essential in ensuring the commitment of mesenchymal stem cell to differentiate into osteoblasts by inhibiting the expression of adipogenic transcription factor while promoting the expression of osteogenic ones. It also enhances osteoblastogenesis in the presence of bone morphogenetic proteins [11].

Apart from mechanical forces, changes in hormone levels, such as estrogen and glucocorticoids, are detected in osteocytes in the interstitial fluid in canaliculi; these changes influence the survival of osteocytes [12,13]. Estrogen plays a crucial role in suppressing bone remodeling activation, primarily by acting on osteocytes. Additionally, it reduces bone resorption by directly influencing osteoclasts or via interactions with osteoblasts/osteocytes and T-cells. Therefore, disruption in the osteocyte network may increase bone fragility [14].

Osteoporosis treatments can be broadly classified into anabolic (examples: romosozumab and teriparatide) and antiresorptive agents (examples: denosumab and bisphosphonates) [15]. These medications can enhance bone mineral density and lower fracture risk but have side effects [16]. Thus, novel bone-protective agents, such as tocotrienol, which is a member of the vitamin E family, have been explored. Structurally, tocotrienol is composed of a chromanol ring and an extended carbon chain containing three double bonds [17], and is classified into four isomers; α-, β-, γ-, and δ-tocotrienol [18]. These isomers coexist in natural sources, such as palm, rice bran, and annatto oils.

Tocotrienol from palm and annatto exerts therapeutic effects on bones through its anti-inflammatory, antioxidant, and mevalonate-suppressive activities [19]. Notably, it is an isoprenoid with 3-hydroxy-3-methyl-glutaryl-coenzyme A reductase-suppressive properties. This action not only lowers production but also regulates the formation and activity of bone cells [20]. It prevents bone loss by promoting osteoblast differentiation and suppressing osteoclast formation [21,22]. The effectiveness of tocotrienol mixtures alone [23,24] or in combination with lovastatin in protecting against bone loss was demonstrated in ovariectomized rats [25]. Tocotrienol facilitates fracture healing in ovariectomized rats [26,27]. A recent study reported that emulsification reduced the amount of palm tocotrienol required to achieve its bone- and joint-protective effects in ovariectomized rats [28].

The regulatory effects of tocotrienol on osteoblasts and osteoclasts have been explored extensively [22,29,30,31]. By contrast, studies on its effects on osteocytes, which are the most abundant bone cells that mediate skeletal remodeling, are relatively limited. Wong et al. (2019) demonstrated that tocotrienol normalized changes in bone-related peptides caused by metabolic syndrome in 12 weeks [32]. In a subsequent study, they showed that annatto tocotrienol reduced empty lacunae but did not alter the number of osteocytes and the expression levels of DMP1 and phosphate-regulating endopeptidase X-linked protein in rats with metabolic syndrome [33]. Mohamad et al. (2021) found that annatto tocotrienol did not alter the osteocyte number and DKK1 protein expression in ovariectomized rats with established bone loss, but reduced the expression of skeletal sclerostin protein in these rats [34]. Given the heterogeneous findings, a valid conclusion on the effects of tocotrienol on osteocytes in animal models could not be drawn.

Therefore, this research sought to investigate the effects of a palm tocotrienol mixture on the osteocyte number and marker expression in ovariectomized rats to explain its bone-protective effects. The study will enhance our understanding of the anti-osteoporotic effects of palm tocotrienol mixtures and facilitate its translation into clinical use.

## 2. Materials and Methods

### 2.1. Ethical Consideration

The animal ethics committee at Universiti Kebangsaan Malaysia granted approval for this study’s protocol (approval code: FAR/FP/UKM/2024/CHIN KOK YONG/29-MAY./1428-JUN.-2024-DIS.-2024). All experimental procedures adhered strictly to the guidelines established by the Malaysian Animal Welfare Act of 2015.

### 2.2. Source and Composition of Palm Tocotrienol

The emulsified (EVNol SupraBio^TM^ 25%) and unemulsified palm tocotrienol (EVNol^TM^ 50%) were supplied by Excelvite Sdn Bhd. (Chemor, Malaysia). The emulsified palm tocotrienol contained 6.9% α-tocopherol, 6.6% α-tocotrienol, 1.1% β-tocotrienol, 9.4% γ-tocotrienol, and 3.2% δ-tocotrienol, and the rest were Labrasol, polysorbate 80, and red palm olein. The unemulsified palm tocotrienol contained 10.3% α-tocopherol, 12.9% α-tocotrienol, 2.0% β-tocotrienol, 19.8% γ-tocotrienol, and 6.3% δ-tocotrienol, and the rest was red palm olein.

### 2.3. Animals and Treatment

A total of 36 adult female Sprague–Dawley rats (three-month-old, 250–300 g) were housed under a controlled environment with a temperature of 25 °C and a cycle of 12 h light and 12 h dark. Unrestricted access to tap water and standard rat chow was provided to the rats. After a seven-day acclimatization period, the rats were randomly divided into six different groups. One group was designated as the baseline group (*n* = 6), while another served as the sham-operated group, receiving refined olive oil orally (*n* = 6). The remaining rats underwent ovariectomy and were further categorized into a negative control group receiving refined olive oil orally (*n* = 6), a group administered with an unemulsified palm tocotrienol mixture (Excelvite, Chemor, Malaysia) at a dosage of 100 mg/kg/day (UPT; orally; *n* = 6), a group given an emulsified palm tocotrienol mixture at 100 mg/kg/day (EPT; orally; *n* = 6), and a group treated with glucosamine sulfate (GS; Mylan Healthcare, Petaling Jaya, Malaysia; 250 mg/kg/day; orally) combined with calcium carbonate (Ca; Bendosen, Batu Caves, Malaysia; 1% in drinking water; *n* = 6). The dosages of UPT, EPT, and glucosamine sulfate with calcium carbonate were selected for their efficacy in preventing osteoporosis [28]. The baseline group was euthanized without any intervention. All groups, except the sham group, underwent bilateral ovariectomy under anesthesia using ketamine (Troy Laboratories, Glendenning, Australia) and xylazine (Indian Immunologicals, Hyderabad, India) at a dose of 0.1 mL/100 g body weight, administered intraperitoneally. The sham group underwent a laparotomy but retained their ovaries. After a recovery period of seven days, the rats received their respective treatments for a duration of 10 weeks. At the conclusion of the treatment period, the rats were humanely euthanized, and their bones were collected for analysis. Blood samples were processed using a Heraeus-Labofuge-400 centrifuge (Thermo Fisher Scientific, Waltham, MA, USA) at 3000 rpm for 10 min to separate the serum, which was subsequently stored at −70 °C until further analysis.

### 2.4. Measurement of Osteocyte Parameters

After removing the soft tissue, the left femur was longitudinally bisected. One half of the bone underwent decalcification at ambient temperature for 30 days using a 10% ethylenediaminetetraacetic acid solution (Sigma-Aldrich, St Louis, MO, USA). Post-decalcification, the femur was embedded in paraffin (Leica Biosystems, Nussloch, Germany) and sectioned into 5 μm slices using a microtome (Leica RM2235, Nussloch, Germany). These sections were subsequently washed with xylene (Merck, Darmstadt, Germany) and rehydrated through a series of graded alcohol concentrations to eliminate the paraffin. Hematoxylin and eosin staining (Abcam, Cambridge, UK) was then applied to the rehydrated sections. The sections were dehydrated again using progressively higher concentrations of alcohol (Merck) followed by xylene. A light microscope (Primostar, Zeiss, Oberkochen, Germany) was employed to examine the slides, focusing on the metaphyseal region located 1 mm below the epiphyseal plate. Images were captured at 100× magnification using Zen 2.6 lite software (Zeiss, Oberkochen, Germany). The number of lacunae and osteocytes in the trabecular and cortical bones was manually counted. These values were normalized by the total bone area assessed, as determined by ImageJ version 1.54m (National Institutes of Health, Bethesda, MD, USA). The calculated indices included osteocyte number per bone area, empty lacunae number per bone area, total lacunae number per bone area, and the percentage of empty lacunae. Three non-overlapping visual fields were examined per section. Three sections were produced for each animal.

### 2.5. Gene Expression

The mRNA levels of *Sost*, *Dkk-1*, and *Dmp-1* were evaluated using real-time reverse transcription polymerase chain reaction (RT-PCR). The fourth lumbar vertebrae from the rats were homogenized in liquid nitrogen. Total RNA isolation from the bone samples was performed with Trizol (Invitrogen, Waltham, MA, USA). The RNA concentration was quantified using a NanoDrop spectrophotometer (Thermo Fisher Scientific, Waltham, MA, USA) at 260 nm. This RNA was then converted into complementary DNA (cDNA) using a OneScript Plus cDNA synthesis kit (Applied Biological Materials Inc., Vancouver, BC, Canada). The reaction mixture was incubated in a thermocycler at 55 °C for 15 min, followed by a 5 s inactivation of the reverse transcriptase at 85 °C.

For the RT-PCR procedure, BlasTaq 2X qPCR MasterMix (Applied Biological Materials Inc., Vancouver, Canada) containing cDNA sample, forward and reverse primers (as specified in Table 1, Integrated DNA Technologies, Coralville, IA, USA), BlasTaq 2X qPCR MasterMix, and nuclease-free water (Vivantis Technologies, Shah Alam, Malaysia) was used. The final reaction volume was 20 µL. The samples were processed using an MIc PCR System (Bio Molecular Systems, Queensland, Australia). The cycling conditions began with an initial denaturation at 95 °C for 3 min, followed by 45 cycles of denaturation at 95 °C for 3 s and annealing/extension at 60 °C for 30 s.

Relative gene expression was quantified using the 2^−ΔΔCT^ method. Cycle threshold (CT) values were obtained from real-time PCR. First, the CT values for the gene of interest were normalized to the reference gene, beta-actin, using the formula ΔCT = CT _gene of interest_ − CT _beta-actin_. Subsequently, the ΔCT values of the treatment groups were normalized to the sham control group using the formula ΔΔCT = ΔCT _treatment group_ − mean ΔCT _sham group_. Finally, the relative gene expression was calculated as 2^−ΔΔCT^, representing the fold change in gene expression relative to the sham group.

### 2.6. Measurement of Serum and Skeletal Redox Status

Venous blood was collected from the rats through cardiac puncture during sacrifice. The blood samples were centrifuged (Heraeus-Labofuge-400, Thermo Fisher Scientific) at 3000 rpm and 4 °C for 10 min. The extracted serum was stored at −70 °C until it was utilized. The fifth lumbar vertebrae of the rats were homogenized in liquid nitrogen and subsequently in phosphate-buffered saline.

The total antioxidant capacity of the samples was determined using a commercialized microtiter plate assay (Catalog No: E-BC-K136-S, Elabscience, Houston, TX, USA) on the basis of the reduction of iron ions. The 8-isoprostane level was determined using an enzyme-linked immunoassay kit (Catalog No: ELK9175, ELK Biotechnology, Denver, CO, USA) according to the manufacturer’s instructions.

### 2.7. Statistical Analysis

The Shapiro–Wilk test was used to determine data distribution. After ensuring a normal distribution, intergroup comparisons were carried out using one-way analysis of variance (ANOVA). If ANOVA was significant, multiple pairwise comparisons were conducted with the Tukey post hoc test. SPSS version 26 (IBM, Armonk, NY, USA) was used for the analyses. A significance threshold was fixed at *p* < 0.05.

## 3. Results

### 3.1. Osteocyte Parameters

Bone histomorphometry analysis revealed no significant differences in the number of trabecular osteocytes, the number of empty lacunae, the number of total lacunae, and the percentage of empty lacunae between the sham and ovariectomized groups (*p* > 0.05). The treatment groups had a significantly higher number of total lacunae and number of trabecular osteocytes than the ovariectomized group (*p* < 0.05). However, the number of empty lacunae and the percentage of empty lacunae did not show significant differences between the treatment and ovariectomized groups (*p* > 0.05; Figure 1).

For the cortical bone, the baseline group had more osteocytes and empty lacunae than the sham group (*p* < 0.05). Ovariectomy did not significantly alter any osteocyte indices in the cortical bone (*p* > 0.05 vs. sham). UPT and Ca + GS significantly increased the number of osteocytes, and all treatments significantly increased the number of total lacunae in the ovariectomized rats (*p* < 0.05 vs. ovariectomized control). The number of empty lacunae and the percentage of empty lacunae did not show significant differences between the treatment and ovariectomized groups (*p* > 0.05; Figure 2).

### 3.2. mRNA Expression Levels of Osteocyte Markers

*Dkk1* mRNA expression significantly increased in the sham group compared to the baseline (*p* < 0.01), while no significant difference in *Dmp1* and *Sost* mRNA expression was noted between them (*p* > 0.05). Ovariectomy significantly increased *Dmp1* (vs. baseline and sham) and *Sost* mRNA expression levels (vs. baseline) (*p* < 0.05). All treatments significantly reduced the *Dmp1* mRNA expression levels, but only Ca + GS reduced *Sost* mRNA expression (*p* < 0.05). *Dkk1* mRNA expression increased after ovariectomy (*p* < 0.05), but none of the treatments reversed the changes (*p* > 0.05) (Figure 3).

### 3.3. Redox Status

Ovariectomy and treatment did not alter the serum and skeletal antioxidant capacity of the rats (Figure 4B–D; *p* > 0.05). Similarly, the skeletal 8-isoprostane level did not change during ovariectomy and treatment (Figure 4C; *p* > 0.05). However, serum 8-isoprostane level was significantly higher in the ovariectomized rats fed with UPT than in the sham and ovariectomized group fed with calcium and glucosamine sulfate (Figure 4A; *p* < 0.05).

## 4. Discussion

This study showed that ovariectomy did not alter trabecular and cortical osteocyte histomorphometric indices but increased *Sost* and *Dmp1* mRNA expression levels. EPT increased the number of trabecular osteocytes, and UPT significantly increased the number of cortical osteocytes in the ovariectomized rats. Both palm tocotrienol formulations reduced *Dmp1* mRNA expression levels. Calcium carbonate, in combination with glucosamine, exerted similar skeletal effects but reduced *Sost* mRNA expression. Changes in circulating and skeletal redox status during ovariectomy and treatment were not specific, and ovariectomized rats treated with UPT displayed higher 8-isoprostane levels than the sham group.

As demonstrated in this study, the effects of ovariectomy on the number of trabecular and cortical osteocytes, empty lacunae, and total lacunae were not significant. Similarly, a study demonstrated that ovariectomy did not affect lacunar porosity and lacunar density after 4 and 14 weeks in Wistar rats [38]. However, our observation is inconsistent with the findings reported by Mohamad et al. [34], who demonstrated that the number of osteocytes significantly decreased in 12-month-old rats four months after ovariectomy [34]. This discrepancy can be due to the relatively young age of our rats. In female mice, multiple studies have highlighted senescent changes in osteocytes as the mice age [39]. Thus, osteocytes in older rodents might be more sensitive to the harmful effects of estrogen deficiency.

Palm tocotrienol supplementation significantly increased trabecular and cortical osteocytes and total lacunae. Mohamad et al. [34] reported a significant increase in osteocytes in ovariectomized rats treated with annatto tocotrienol [34]. The current study did not show any significant changes in the number or percentage of empty lacunae in the trabecular bone, but the previous literature reported otherwise. Wong et al. [33] reported that tocotrienol treatment prevented the increase in empty lacunae in rats fed with a high-carbohydrate high-fat diet, probably because of the anti-inflammatory and antioxidative properties of tocotrienol [33]. The discrepancy between these findings and other studies might be due to differing experimental conditions, such as the method of inducing bone loss (ovariectomy versus high-carbohydrate high-fat diet), and the sex of the rats used. The effect of tocotrienol could be enhanced by a high-carbohydrate and high-fat diet model, in which oxidative stress and inflammation are more evident than those in the ovariectomized model used in this study.

DMP1 is crucial to the regulation of bone matrix mineralization [40]. SOST and DKK1 are the inhibitors of the Wnt pathway, suppressing osteoblast activity and bone formation [41]. In the present study, ovariectomy increased *Dmp1* mRNA expression levels relative to those of the sham group and increased *Sost* mRNA expression compared with the baseline. The increased mRNA expression levels of these markers may be attributed to the high bone remodeling typically observed with estrogen deficiency. Increased SOST can suppress bone formation [42]. The downregulation of *Dkk1* mRNA expression with ovariectomy observed in this study can be due to the increase in *Sost* mRNA expression. A study found a negative correlation between SOST and DKK1 expression, suggesting a compensatory feedback mechanism in bone formation [43].

Upon treatment with palm tocotrienol, *Dmp1* mRNA expression levels were significantly reduced in the ovariectomized rats. *Sost* expression was decreased but not significantly by tocotrienol supplementation. The decreasing trend of both *Dmp1* and *Sost* mRNA expression is suggestive of the suppression of bone remodeling. The study of Mohamad et al. [34] demonstrated that self-emulsifying annatto tocotrienol reduced SOST levels significantly in ovariectomized rats. A previous study reported a significant reduction in DKK1 protein expression levels upon annatto and palm tocotrienol treatment [32]. However, the current study was unable to replicate the same results. The difference in composition among the tocotrienol mixtures could contribute to this discrepancy. Almost 90% of the total vitamin E content in annatto bean is δ-tocotrienol, whereas only 12% of the total vitamin E content in palm fruit is δ-tocotrienol. A previous in vitro experiment showed that δ-tocotrienol was the most osteogenic compared to the other tocotrienol isomers [44], suggesting that the difference in natural tocotrienol mixture could contribute to the difference in skeletal protective effects.

Oxidative stress is the result of an imbalance between oxidants and antioxidants in the body. 8-Isoprostane, a stable biomarker of oxidative stress, is generated by the nonenzymatic free radical-catalyzed oxidation of arachidonic acid [45]. Total antioxidant capacity is often used in assessing the body’s overall antioxidant status, including enzymes, such as glutathione peroxidase, and nonenzymatic antioxidants, such as reduced glutathione and vitamin E [46]. Estrogen displays direct antioxidant effects by scavenging free radicals and indirectly upregulates antioxidant enzymes. Ovariectomy, which ultimately leads to estrogen deficiency, is expected to increase oxidative products while reducing antioxidant activities in rats [47,48]. However, this trend was not observed in the current study, probably because estrogen is deficient only for a short time.

Tocotrienol is a known antioxidant. Treatment with palm tocotrienol reduces lipid peroxidation product levels and increases glutathione peroxidase activity in the femur of rats. These effects were not observed after a-tocopherol supplementation [49]. Notably, tocotrienol improves antioxidant enzyme levels and activities in animal models of osteoporosis [47]. However, the current study was not able to replicate similar trends.

Several limitations should be highlighted in the current study. The protein translated from the genes was not examined. Given that these proteins can be expressed by cells other than osteocytes [50,51], the localization of the proteins can be achieved by immunostaining. An in vitro study using bone marrow mesenchymal cells treated with tocotrienol could be used to validate the findings of this study. Three-month-old rats were used in this study because they are sexually mature and extensively used as models of osteoporosis. However, the skeletal growth of rats differs from that of humans and can extend beyond 26 weeks of age despite the slower pace [52]. Older rats can more effectively demonstrate the skeletal effects of estrogen deficiency and tocotrienol, but this procedure may have logistical difficulties. Despite these limitations, this study provided novel insights into the osteocyte-regulating activities of palm tocotrienol when estrogen is deficient.

## 5. Conclusions

Palm tocotrienol may exert its bone-protecting effects by preserving the number of osteocytes and modulating the osteocytes’ bone remodeling activities in ovariectomized rats, particularly through modulating *Dmp1* and *Sost* expression. However, based on the differences between our findings and those of previous studies, the action of tocotrienol may be dependent on the animal model, the animals’ age, and the composition of the mixture. Further studies are warranted to illustrate the mechanism by which these variables affect palm tocotrienol’s actions on osteocytes and skeletal redox status.

## Figures and Tables

**Figure 1 biomedicines-13-01220-f001:**
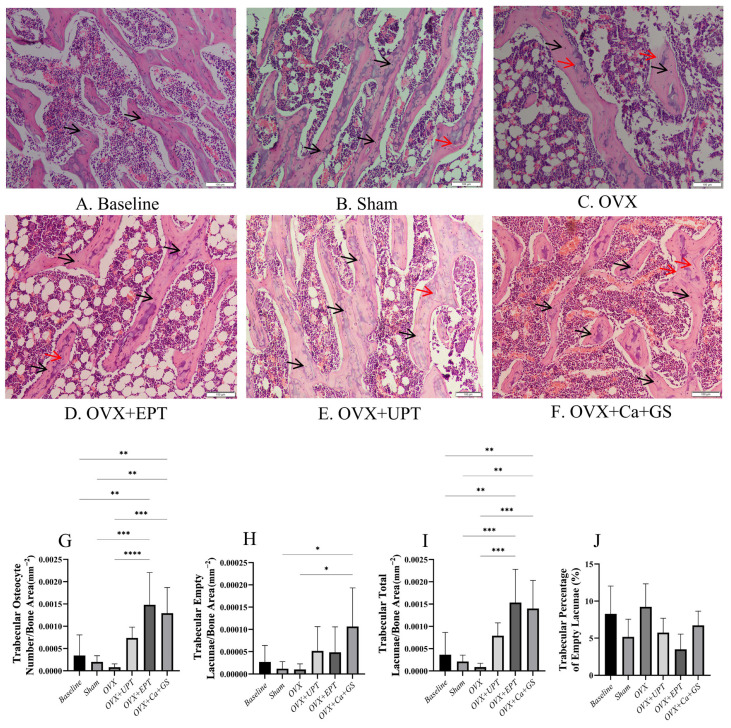
Hematoxylin and eosin-stained femoral sections of trabecular bone (**A**–**F**) were used in quantifying osteocytes (**G**) and empty lacunae (**H**) and determining the total lacunae (**I**) and the percentage of empty lacunae (**J**). Results are reported as the mean value and the error bars indicate the standard deviation, with each group comprising six rats. Abbreviations: OVX, ovariectomized group; OVX + Ca + GS, ovariectomized rats treated with glucosamine sulfate and calcium carbonate; OVX + UPT, ovariectomized rats treated with unemulsified palm tocotrienol; OVX + EPT, ovariectomized rats treated with emulsified palm tocotrienol. Legend: The red arrow shows examples of empty lacunae, whereas the black arrow shows examples of osteocytes. * indicates *p* < 0.05, ** *p* < 0.01, *** *p* < 0.001 and **** *p* < 0.0001.

**Figure 2 biomedicines-13-01220-f002:**
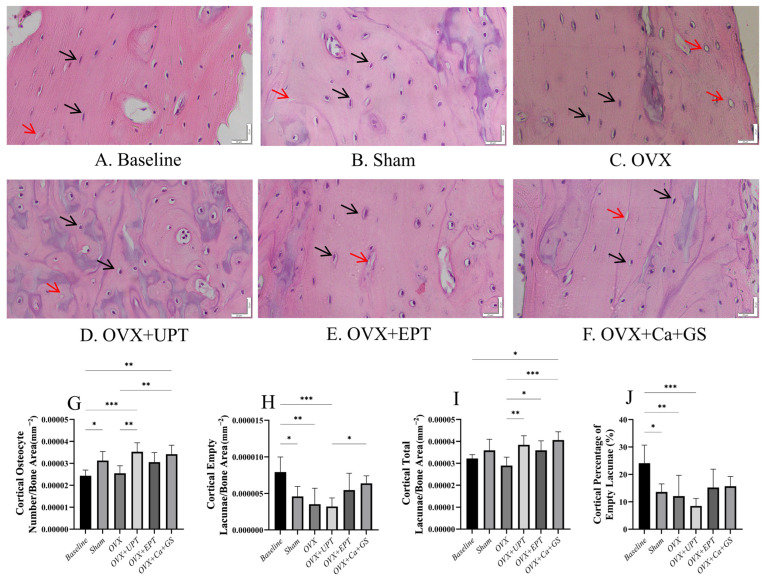
Hematoxylin and eosin–stained femur section histological slides of the cortical bone (**A**–**F**) were used in quantifying osteocytes (**G**) and empty lacunae (**H**) and determining total lacunar count (**I**) and the percentage of empty lacunae (**J**). Results are reported as the mean value and the error bars indicate the standard deviation, with each group comprising six rats. Abbreviations: OVX, ovariectomized group; OVX + Ca + GS, ovariectomized rats treated with glucosamine sulfate and calcium carbonate; OVX + UPT, ovariectomized rats treated with unemulsified palm tocotrienol; OVX + EPT, ovariectomized rats treated with emulsified palm tocotrienol. Legend: The red arrow shows examples of empty lacunae, while the black arrow shows examples of osteocytes. * indicates *p* < 0.05, ** *p* < 0.01, and *** *p* < 0.001.

**Figure 3 biomedicines-13-01220-f003:**
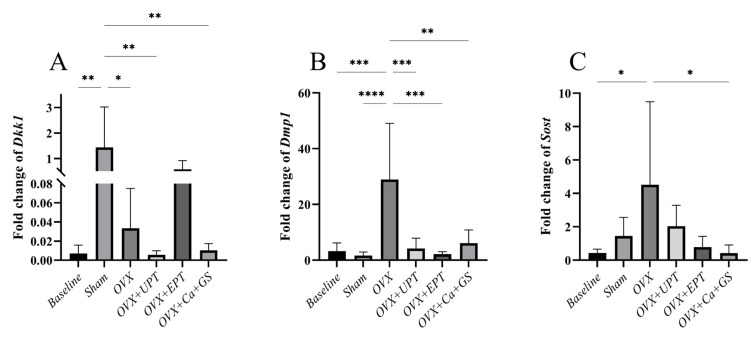
Dickkopf-1 (**A**), dentin matrix protein-1 (**B**), and sclerostin (**C**) mRNA expression levels of each group. Results are reported as the mean value and the error bars indicate the standard deviation, with each group comprising six rats. Abbreviations: OVX, ovariectomized group; OVX + Ca + GS, ovariectomized rats treated with glucosamine sulfate and calcium carbonate; OVX + UPT, ovariectomized rats treated with unemulsified palm tocotrienol; OVX + EPT, ovariectomized rats treated with emulsified palm tocotrienol. Legend: * indicates *p* < 0.05, ** *p* < 0.01, *** *p* < 0.001 and **** *p* < 0.0001.

**Figure 4 biomedicines-13-01220-f004:**
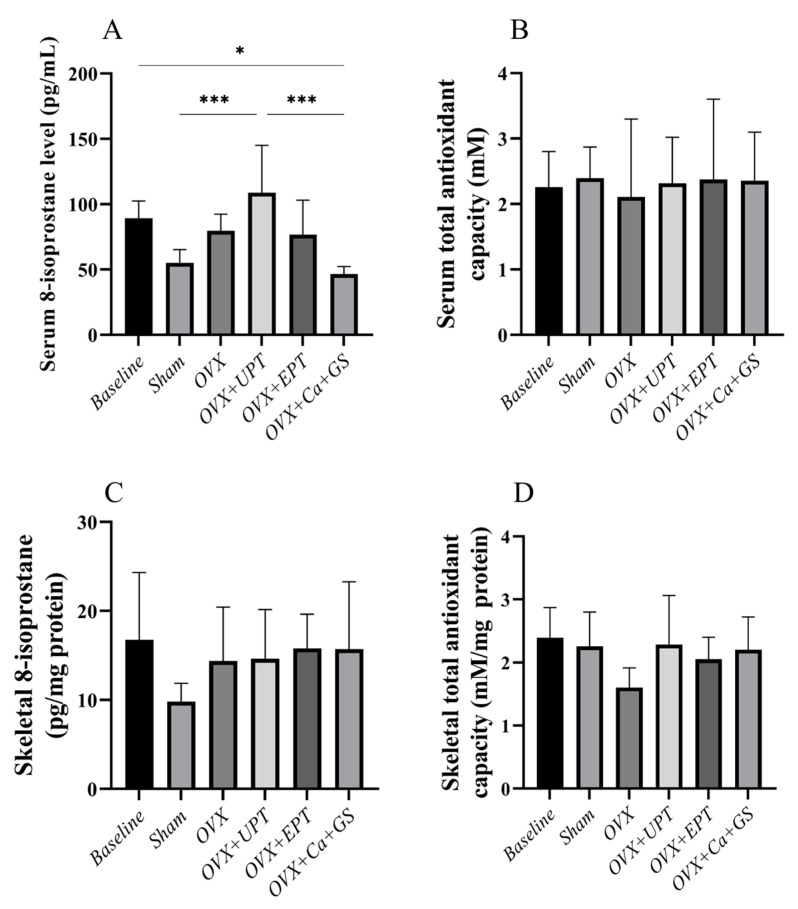
Serum 8-isoprostane (**A**), serum total antioxidant capacity (**B**), skeletal 8-isoprostane (**C**), and skeletal total antioxidant capacity (**D**) of each group. Results are reported as the mean value and the error bars indicate the standard deviation, with each group comprising six rats. Abbreviations: OVX, ovariectomized group; OVX + Ca + GS, ovariectomized rats treated with glucosamine sulfate and calcium carbonate; OVX + UPT, ovariectomized rats treated with unemulsified palm tocotrienol; OVX + EPT, ovariectomized rats treated with emulsified palm tocotrienol. Legend: * indicates *p* < 0.05 and *** *p* < 0.001.

**Table 1 biomedicines-13-01220-t001:** Primer sequence for RT-PCR.

mRNA	NCBI Reference Sequence	Forward	Reverse	Reference
*Dkk1*	NM_001106350	ATGAGGCACGCTATGTGCTG	CTCGAGGTAAATGGCTGTGGTC	[35]
*Sost*	NM_030584.2	TGCTTCGGTGACAGGTAG	CACGTCTTTGGTGTCATAAG	[36]
*Dmp1*	NM_203493.4	ACCTTTGGAGACGAAGACAATGGC	TGTCTTCACTGGACTGTGTGGTG	[37]

## Data Availability

The data that support the findings of this study are available from the corresponding author, K.-Y. Chin, upon reasonable request.

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
