# Peer review of "Palm Tocotrienol Preserves Trabecular Osteocyte Indices and Modulates the Expression of Osteocyte Markers in Ovariectomized Rats"

_biomedicines, 2025, doi:10.3390/biomedicines13051220_

Round 1
Reviewer 1 Report
Comments and Suggestions for Authors
- The introduction does not fully explain the specific molecular mechanism of osteoblasts regulating bone metabolism, and it is suggested to be supplemented.
- The conclusion only detects the mRNA level of DMP1 and SOST. Is it necessary to verify the protein expression using Western blot or immunohistochemistry?
- Palm Tocotrienol did not improve oxidative stress indicators (e.g., total antioxidant capacity), which contradicts previous studies' conclusions; do the conclusions need to explore the reasons? (doi: 10.1152/ajpheart.zh4-0452-retr.2012) (10.1111/j.1742-7843.2008.00241.x)
- Potential mechanisms (e.g. epigenetic modifications) of how Palm Tocotrienol regulates bone mineralisation through DMP1 with no effect on SOST expression are not explained and are suggested to be added to the Discussion.
- The micrographs in Figures 1 and 2 have insufficient resolution and are poorly labelled for scale.
Please check the language of the whole manuscript to make sure the correction of all expressions and spelling of words. Such as: "2.2 Line 2 under controlled environmental"
Author Response
Dear reviewer,
Thank you for the constructive comments. Please see the attachment for our reply. Thank you.

Reviewer 2 Report
Comments and Suggestions for Authors
The study investigates the effects of palm tocotrienol on the quantity of osteocytes and the expression of osteocyte-specific markers in ovariectomized rats, which is of certain significance for revealing the mechanism of Tocotrienol's bone protection. There are many studies on the role of tocotrienols in bone remodeling. While the study addresses a relevant topic, the novelty appears limited. And the selection of 3-month-old animals seems inappropriate. Moreover, detecting the expression of bone biochemical markers at the gene level by RT-PCR alone does not seem to be enough to explain the accuracy of the results. The protein translated from the genes should be examined. I recommend that this manuscript be rejected for publication.
Author Response

(The authors gave the same response as above.)

Reviewer 3 Report
Comments and Suggestions for Authors
Review:
This manuscript investigates the effects of palm-derived tocotrienol supplementation on bone health using an ovariectomized (OVX) rat model. The study is timely and potentially relevant, given the search for novel bone anabolic agents. However, several issues—particularly regarding experimental design interpretation, methodological reporting, and data consistency—need to be addressed for the conclusions to be supported.
Positive attributes:
- Novel potential bone protective agent: The paper addresses a growing need in the field by attempting to test bone protective effects of palm tocotrienol.
- Good experimental controls: The authors report baseline and sham and OVX controls for the experiment.
Negative attributes:
Minor:
- Section 2.2 wording:
“After a seven-day acclimatization period, the rats were randomly divided into six distinct groups.”
If groups were randomized, describing them as "distinct" may imply predefined characteristics and introduce the perception of selection bias. Consider revising to:
“...randomly divided into six different groups.”
- Introduction, paragraph 4:
The sentence, “Thus, novel bone-protective agents, such as tocotrienol, which is a member of vitamin E, have been explored.” has two issues-
- A more accurate phrasing would be:
“...a member of the vitamin E family” or “...a member of the vitamin E family of diterpenoids.”
- Tocotrienol's role as a “bone-protective” agent is still under investigation and should be presented with appropriate caution, such as:
“...potential bone-protective properties...” - Isomer-specific effects of tocotrienol:
References 22–24 represent different tocotrienol isomer compositions (e.g., delta-rich vs. alpha/gamma-rich). The authors should:
- Report the composition of the tocotrienol mixture used in their own study.
- Discuss potential differences in effects across isomers, especially since their results diverge from some earlier studies.
- Citation style consistency:
In the Discussion section, references such as 32 and 33 can be simplified to “Wong et al., 2022” and “Mohamad et al., 2021” instead of listing all author names, unless journal guidelines require full names.
Major:
- Animal housing:
The authors describe use of “controlled environmental with a temperature of 27°C and a cycle of 16 hours light and 8 hours dark.” which is uncommon with rodent studies. A housing temperature of 22-24C and 12:12 light:dark cycle is the common practice. The authors should justify the use of their set-up and clarify if they align with the institution's animal care guidelines.
- Representative images and quantification mismatch:
There appears to be a disconnect between Figures 1C/E and the quantification in 1G. For example, the increase in trabecular osteocyte count (~0.0002 for OVX to ~0.0015/mm² for OVX EPT) is not visually supported by the provided images. The authors should clarify the method of selecting regions of interest for quantification. Images should be verified as representative of group averages.
- Lack of replication and ambiguous interpretation:
While the study did not replicate some previous findings, the conclusion lacks nuance. It should be made clear whether this reflects: A true limitation of tocotrienol in certain contexts, or a result of specific variables in the current study (e.g., animal age, formulation, model type, duration, etc.).
- Insufficient evidence for osteocyte number change:
Counting osteocytes from paraffin sections, without corroborating markers or 3D structural analysis, may not suffice to draw conclusions about osteocyte dynamics. Consider adding_
Dynamic histomorphometry or in situ hybridization,
Immunohistochemistry for osteocyte markers (e.g., DMP1, SOST)
Additional recommendation:
- Ex vivo/in vitro experiments:
Testing tocotrienol effects on primary osteoblasts and osteoclasts (e.g., using bone marrow-derived cells) could complement the current in vivo data and help dissect mechanisms of action. - Bone microarchitecture:
DXA or microCT analysis would greatly enhance the dataset and provide structural insights beyond histology. - Dynamic histomorphometry:
Inclusion of calcein or alizarin red labeling would allow assessment of bone formation rate and mineral apposition rate, providing critical context to osteocyte number data.
Author Response

(The authors gave the same response as above.)

Reviewer 4 Report
Comments and Suggestions for Authors
The problem of bone diseases, especially osteoporosis, is acute today. Effective and safe pharmacological prevention and control of negative age-related changes in bone tissue is an important task of modern medicine. The work of the authors is devoted to the actual problem and the experiment is very well planned. At the same time, the paper has serious shortcomings that do not allow it to be accepted for publication in its present form.
- Keywords do not match the content of the MS. The authors did not assess the general condition of the bone tissue, did not look at the parameters of the bone matrix, the bone base, but only at the number of lacunae. This is not an assessment of bone remodelling. The authors did not look at components of WNT-signaling.
- Sections 2.3 do not specify how many visual fields were examined in each biological sample.
- In section 2.4. "Measurement of bone biochemical markers" describes the assessment of differential expression of genes involved in the regulation of bone homeostasis, not the measurement of biochemical markers. Title does not correspond to content. Why was normalisation compared to sham control? Why were baseline and sham not compared if significant differences were observed?
- Figures 1 and 2 are of poor quality. Low resolution of microphotographs, bars difficult to distinguish. Differences between groups are difficult to interpret from the graphs. In Figure 1, the scatter in the baseline group exceeds the values themselves. It is necessary to improve the quality of the micrographs, add adequate bars, make the comparison groups clearer, explain the magnitude of the scatter or make additional measurements.
- Section 3.2 should be completely rewritten, as there are significant differences in the expression of the genes studied between the baseline and sham groups. At the same time, the authors define delta CT only in comparison with the sham group, without explaining or discussing the differences in the control groups. As presented, the results do not stand up to criticism.
- The authors refer mainly to data obtained by their team when discussing their results. No other actual data are cited. This significantly reduces the quality and reliability of the results obtained. Alternative sources should be provided.
- Reference 44 does not match the claim.
- In "conclusion", the authors draw a deduction about the effect on Wnt signaling. which is inadmissible, because the authors have not conducted any research on the components of the signaling pathway.
Author Response

(The authors gave the same response as above.)

Round 2
Reviewer 3 Report
Comments and Suggestions for Authors
Authors response to Major point no.2:
|
Representative images and quantification mismatch: There appears to be a disconnect between Figures 1C/E and the quantification in 1G. For example, the increase in trabecular osteocyte count (~0.0002 for OVX to ~0.0015/mm² for OVX EPT) is not visually supported by the provided images. The authors should clarify the method of selecting regions of interest for quantification. Images should be verified as representative of group averages. |
Thank you for the comment. We request the reviewer to reconsider this comment because the data presented in Figure 1G are osteocyte number/bone area, not osteocyte number per se. The trabecular bone area should be accounted for when generating the quantitative data, not just based on the osteocyte quantity on the image. We have clarified in the methods that the region of interest was the metaphyseal region located 1 mm below the epiphyseal plate (Line 148-149), and three non-overlapping visual fields per section were examined. Three sections were produced for each animal (Line 155-156).
|
Reviewer comment:
1. The clarification regarding the quantification being normalized to bone area is appreciated. However, the original concern was not with the normalization method per se, but with the apparent disconnect between the representative images (Figures 1C/E) and the magnitude of change reported in Figure 1G.
To improve clarity and alignment between the representative images and the quantified data, it would be helpful for the authors confirm that they are representative of group-level trends. Inclusion of annotated images or supplementary figures that better reflect the group averages could help readers better appreciate the reported differences.
- The authors have responded to the remaining comments in a clear and satisfactory manner.
Additional comment:
In the revised version, some figures (notably Figures 1 and 2) appear rotated, which may affect readability—particularly panels 1G–J and 2G–J. Adjusting the orientation could improve clarity for the reader.
Author Response
Comment 1:
1. The clarification regarding the quantification being normalized to bone area is appreciated. However, the original concern was not with the normalization method per se, but with the apparent disconnect between the representative images (Figures 1C/E) and the magnitude of change reported in Figure 1G.
To improve clarity and alignment between the representative images and the quantified data, it would be helpful for the authors confirm that they are representative of group-level trends. Inclusion of annotated images or supplementary figures that better reflect the group averages could help readers better appreciate the reported differences.
Reply 1:
Thank you for your comment. We have changed the images to better reflect the magnitude of changes in Figure 1.
Comment 2:
In the revised version, some figures (notably Figures 1 and 2) appear rotated, which may affect readability—particularly panels 1G–J and 2G–J. Adjusting the orientation could improve clarity for the reader.
Reply 2:
Thank you for the suggestion. We have rotated Figures 1 and 2 as per your suggestion.
Reviewer 4 Report
Comments and Suggestions for Authors
After revision, the manuscript was considerably improved. However, the author did not correct one small error: in Figures 1 and 2, the value of the scale bar in the micrograph of bone sections was completely illegible, which should be corrected as it gives the impression that the images shown were taken at different magnifications.
Author Response
Comment 1:
After revision, the manuscript was considerably improved. However, the author did not correct one small error: in Figures 1 and 2, the value of the scale bar in the micrograph of bone sections was completely illegible, which should be corrected as it gives the impression that the images shown were taken at different magnifications.
Reply 1:
Thank you for your comment. We have carefully checked and improved the resolution of Figures 1 and 2, so that the scale bars are visible.